# Clinical Trials of Cellular Therapies in Solid Tumors

**DOI:** 10.3390/cancers15143667

**Published:** 2023-07-19

**Authors:** Simona Secondino, Costanza Canino, Domiziana Alaimo, Marta Muzzana, Giulia Galli, Sabrina Borgetto, Sabrina Basso, Jessica Bagnarino, Chiara Pulvirenti, Patrizia Comoli, Paolo Pedrazzoli

**Affiliations:** 1Oncology Department, Fondazione IRCCS Policlinico San Matteo, 27100 Pavia, Italy; domiziana.alaimo01@universitadipavia.it (D.A.); marta.muzzana01@universitadipavia.it (M.M.); gi.galli@smatteo.pv.it (G.G.); sabrina.borgetto01@universitadipavia.it (S.B.); p.pedrazzoli@smatteo.pv.it (P.P.); 2Department of Internal Medicine and Medical Therapy, University of Pavia, 27100 Pavia, Italy; costanza.canino@gmail.com; 3Cell Factory, Fondazione IRCCS Policlinico San Matteo, 27100 Pavia, Italy; s.basso@smatteo.pv.it (S.B.); j.bagnarino@smatteo.pv.it (J.B.); c.pulvirenti@smatteo.pv.it (C.P.); p.comoli@smatteo.pv.it (P.C.); 4Pediatric Oncoematology Unit, Fondazione IRCCS Policlinico San Matteo, 27100 Pavia, Italy

**Keywords:** immunotherapy, adoptive T cell therapy, ACT, CAR-T, solid tumors

## Abstract

**Simple Summary:**

Cell therapy approaches, including adoptive cell therapy with engineered T cells remains challenging in the clinical setting of solid tumors. Clinical results have not been encouraging so far, with a general lack of significant therapeutic response and presence of on-target off-tumor toxicity. For this reason, novel cell therapy programs are currently exploring (i) advanced therapy medicinal products capable of increasing T cell affinity or avidity; and (ii) cell therapy in combination with other therapeutic agents. Our review will focus on the current clinical research in this setting, which will likely play a role in improving cancer treatment outcomes in the near future.

**Abstract:**

In the past years cancer treatments have drastically changed, mainly due to the development of immune checkpoint inhibitors capable of immune modulation in vivo, thus providing major clinical benefit in a number of malignancies. Simultaneously, considerable technical refinements have opened new prospects for the development of immune cell-based medicinal products and unprecedented success with chimeric antigen receptor (CAR)-T cells targeting B-cell hematologic malignancies has been obtained. However, T cell therapies introduced and performed in the field of solid tumors have produced so far only limited responses in selected patient populations. This standstill is attributable to the difficulty in identifying target antigens which are homogeneously expressed by all tumor cells while absent from normal tissues, and the limited T cell persistence and proliferation in a hostile tumor microenvironment that favors immune escape. Replicating the results observed in hematology is a major scientific challenge in solid tumors, and ongoing translational and clinical research is focused on obtaining insight into the mechanisms of tumor recognition and evasion, and how to improve the efficacy of cellular therapies, also combining them with immune checkpoint inhibitors or other agents targeting either the cancer cell or the tumor environment. This paper provides an overview of current adaptive T cell therapy approaches in solid tumors, the research performed to increase their efficacy and safety, and results from ongoing clinical trials.

## 1. Introduction

The past decades have seen an increasing activity among different kinds of immunotherapy in cancer patients. The first experiences were reported in the 1980s, and concerned the development of IL-2 for the treatment of melanoma and renal cell carcinoma [1,2].

Following up approaches to immunotherapy other than IL-2 did not have significant success, and adoptive T cell therapy (ACT), despite a number of early phase I-II trials, did not become part of standard care in the management of solid tumors. Multiple cancer mechanisms have been identified with the aim of escaping immune recognition, including the downregulation of tumor antigens, the generation of a microenvironment able to escape the immune system, and the secretion of cytokines and negative immune regulators thought to be able to silence immune effectors [3,4].

Nevertheless, considerable progress has been made in the field of cancer immune evasion and host–tumor interaction, which has developed immune checkpoint inhibitors [5] and antibodies against programmed death (PD) or programmed death-ligand (PD-L), with the aim of increasing the immune response against tumor cells [6].

On the other hand, a new scenario has opened for the development of immune-cell-based therapy from the success of chimeric antigen receptor (CAR)-T cells against hematologic malignancies [7,8,9]. CARs are chimeric immunoglobulin T cell receptor (TCR) molecules derived from transgenes encoding for single-chain variable fragments, which originate from antibodies capable of recognizing tumor-associated antigens (TAA) [10,11]. Activation signals induce CAR-T cells, leading to cytokine release and transcription factor expression, to promote T cell survival and function and cytotoxic activity against cancer cells [12].

Whereas CAR-T trials for the treatment of leukemia and lymphoma have shown durable clinical remission of the disease [9,13], CAR-T therapy targeting solid tumors has been disappointing. Among the hurdles identified has been the difficulty of recognizing target tumor antigens, and of trafficking T cells to the tumor site, avoiding a hostile tumor microenvironment. It has been suggested that control of advanced solid tumors will not be reached with a single therapeutic drug, but rather with combinations of either conventional or immunotherapy treatments [14]. Concerning this, immune checkpoint inhibitors, such as the CTLA4 antibodies and the anti PD1-PDL1, may have a role in increasing the response to ACT.

Here, we discuss ACT in solid tumors in clinical development, and consider the challenges plaguing the field, with the aim of succeeding in immune evasion to pave the way for effective tumor control.

## 2. Adaptive Cell Therapy with T Cells Generated and Expanded In Vitro

T cells have a critical role in immune surveillance for cancer, and the lymphocytes infiltrating the tumor‘s environment also have a favorable prognostic role.

First attempts were performed for advanced renal cell carcinoma and metastatic melanoma: recombinant human interleukin-2 (rhIL-2) used in these settings, was capable of favoring human T cell growth [1,2]. This encouraged the collection of T cells generated from patients’ lymphocytes and expanded in vitro with high-dose rhIL-2. The authors reported objective clinical responses, and provided the clinical use of T cells derived from in vitro culture.

### 2.1. Tumor-Infiltrating Lymphocytes (TILs)

Despite the encouraging responses reported in melanoma [1,2,15], T cells isolated from tumor biopsies (TILs) did not obtain the same results in other cancers.

Early TIL trials reported responses in about 50% of melanoma patients [15,16], and pretreatment with lymphodepleting chemotherapy led to improved TIL persistence in vivo, generating an immune environment to expand the T cells infused [16]. After the first trial that had seen the use of cyclophosphamide prior to T cell infusion [1], subsequent trials were designed with the aim of improving results, through the combining of fludarabine with cyclophosphamide, and testing different doses of total body irradiation [16,17].

Interesting data were reported in hepatocellular carcinoma (HCC) patients, who were randomized to receive adoptive immunotherapy (*n* = 76) or no adjuvant treatment, (*n* = 74) after surgical resection [18]. Cellular therapy decreased the frequency of recurrence by 18% compared with controls, and reduced the risk of recurrence by 41%. Therefore, it is encouraging that TILs could suppress tumor growth, supporting the issue of the major benefit of T cell therapy in patients with a minor burden of disease. Unfortunately, TILs became exhausted over time, because immune checkpoint inhibitor molecules, such as PD-1, are up-regulated in TILs isolated from HCC specimens [19]. Moreover, the isolation of an adequate number of TILs from HCC tissue, and the lack of immunogenic antigens, add additional barrier to using TILs as an effective therapy for HCC patients [19].

More recently, selected autologous TILs have shown activity in several other epithelial malignancies, such as cholangiocarcinoma, gynecological (both cervical and ovarian), lung, colorectal, and breast cancer [20,21,22,23,24,25]. Despite the growing interest in T cell therapy, the clinical use of TILs in advanced epithelial cancer has demonstrated limited responses [26]. An important factor may be the poor efficacy of TIL induced by the suppressive tumor microenvironment [27]. An alternative strategy for producing effective ACT therapies is to generate neoantigen-specific T cells de novo by means of TCR-engineered T cells [27]. Initial studies for screening neo-antigens utilized autologous tumor cell c-DNA libraries [28,29,30], but these approaches involved a laborious process for identifying neoepitopes. Subsequently, advances in next-generation sequencing allowed for the recognition of immunogenic tumor antigens though in silico analysis. Using a major histocompatibility complex-binding algorithm, putative T cell epitopes were identified and synthesized and then assessed for recognition by TILs [31].

Currently, there are many TIL ACT clinical studies recruiting patients with gynecological cancer (NCT04611126; 0310845), urothelial carcinoma (NCT04383067), breast cancer (NCT04111510), lung cancer (NCT04072263; NCT03853187; NCT02133196), biliary tract cancer (NCT03801083), and mesenchymal tumors (NCT03449108), as reported in Table 1.

### 2.2. Tumor-Infiltrating Lymphocytes (TILs) in Combination with Immune Checkpoint Inhibitors (ICIs)

Among immunotherapy in solid tumors, immune checkpoint inhibitors (ICIs) have obtained interesting results. Melanoma is one of the malignancies that is most responsive to ICIs, and the first results concern anti CTLA-4 (Ipilimumab), which can break peripheral tolerance to self-tissues and induce an antitumor response, reporting a significant improvement in overall survival with a risk reduction of 32–35% [32].

PD-1 and PD-L1 stand out amongst the cancer immunotherapies that have been approved for clinical use, for different diseases [6,33,34,35]. The interaction between programmed death ligand-1 (PD-L1) and its receptor PD-1 has a role in immune self-tolerance and it is typical for the tumor micro-environment to be enriched in PD-L1 in order to promote tumor tolerance by the immune T-cells. Overexpression of PD-L1 has been demonstrated to inhibit the T cell-mediated antitumor response, favoring tumor escape [27]. The combination of different immune therapies has been the following step to improve the clinical response. In this way, the combination of Nivolumab and Ipilimumab in melanoma patients reported a significant increase in clinical efficacy, with a median progression-free survival of 11.5 months versus 6.9 in favor of the combo regimen [33]. While Ipilimumab can activate T cells, the anti-PD-1 antibodies block PD-1 on the activated T cells that are inhibited by PD-L1 expression of cancer cells [36].

In the tumor microenvironment, the high levels of PD-L1 are able to inhibit the activity of transferred TILs, blocking the PD-1 receptor. Therefore, one hypothesis for overcoming these hurdles is to merge TILs with anti PD-1 therapy. One interesting study in this setting concerned the combination of ICIs with TILs in ovarian cancer [37]. Although a modest clinical benefit was observed, interesting immunological observations were reported. The preconditioning with the anti-CTLA-4 antibody Ipilimumab was employed to increase tumor infiltration of tumor-reactive lymphocytes before tumor tissue harvest for TIL production, and data reported an increase in the success rate of ex vivo TIL expansion and an improved quality of the TIL product [37]. These findings confirmed previous animal data, suggesting that, if there is an endogenous antitumor immune response in the animals after tumor implantation, CTLA4 blockade could enhance that endogenous response, and induce tumor regression [36].

Also, the association between TIL and the PD-1 blockade could represent an efficacy scenario for testing TIL. Some experiences have documented a clinical response to TIL after relapse following anti PD-1 therapy, as reported for lung cancer [38]. PD-1 regulates effector T cells within the tissue and tumor microenvironment, enhancing the activity of effector T cells and NK cells. In addition, chronic antigen exposure, appearing in chronic disease as cancer, can manage high levels of PD-1 expression, which can promote a state of exhaustion [36]. As reported, PD-1 is also expressed in a large proportion of TILs from different tumor types [39], and increased levels of PD-1 expression may reflect an exhausted state. On the other hand, PD-1 ligands are commonly upregulated on the tumor cell surface of many different tumors. These findings provide the basis for a combination of TILs and anti-PD-1 immunotherapy, with the aim of enhancing antitumor functions in the tumor microenvironment. Combined TIL and anti-PD-1 therapy reported clinical benefit in the treatment of chemotherapy-resistant advanced osteosarcoma [40,41]. Notably, this strategy displayed an interesting antitumor effect and objective response, with 22 out of the 60 (36.6%) patients enrolled [40], displaying clinical tumor regression and reporting a median overall survival (mOS) of 13.6 months [40], whereas the mOS of patients with refractory disease receiving chemotherapy is no more than 6 months [42]. Similar results were obtained in 30 patients receiving TILs plus nivolumab, and reporting a 33% response rate [41]. To date, several studies have analyzed the use of single anti-PD-1 therapy against osteosarcoma, with an overall response rate of 5% [41,43]. These findings suggest that single-agent anti-PD-1 therapy may not be an effective treatment strategy, while the combination therapy may be a potential strategy for improving treatment of metastatic osteosarcoma. Moreover, a subpopulation of PD-1^+^T lymphocytes was noticed in the cultured TILs [40,41], suggesting that a PD-1 blockade may boost the cytotoxicity of TILs.

Up to the present date, several studies are ongoing with the aim of exploring TILs in combination with immune checkpoint inhibitors (Table 1).

### 2.3. Virus-Specific T Cells as ACT

The immune system has developed to recognize viral antigens; therefore, virally infected tumors such as cancers associated with the human papillomavirus (HPV) or Epstein–Barr virus (EBV) can trigger T cell immunity [44].

EBV-specific cytotoxic T cells (CTLs) expanded from the hematopoietic stem cell donor have been shown to prevent and treat EBV-associated lymphoproliferative disease [45,46,47]. Subsequently, autologous EBV-specific CTLs have been successfully employed in the treatment of EBV-associated cancers, such as nasopharyngeal carcinoma (NPC) [48,49,50,51]. These trials have so far documented an encouraging 20% objective response rate, with a 10% rate of complete response. With the aim of increasing clinical response and enhancing T cell survival, lymphodepleting chemotherapy has been administered, based on previous melanoma TILs therapy, without improving outcomes [50,51]. Authors hypothesized that the profound lymphopenia may have also depleted those factors able to promote the T cell effector function or expansion [51], and the presence of CD4+ or other immune T cells are also important for promoting and enhancing T cell effector activity against tumor cells [21,27].

HPV-TILs have also been documented as having a role in the regression of HPV-related cancers [20,52]. Among 29 patients treated with HPV-TILs, 25% reported an objective tumor response; moreover, 5 out of 18 patients (28%) in the cervical cancer cohort and 2 out of 11 (18%) in the non-cervical cancer cohort reported the same [52]. Interestingly, patients recording the highest frequency of HPV-reactive T cells in their peripheral blood (PB) after therapy reported objective tumor responses [20]. Moreover, responding patients who recorded elevated PB T cell reactivity against E6 and/or E7 reported prolonged repopulation with the infused HPV-reactive T cells [20].

Albeit the response rate reported is modest, it may reflect in part the mechanism of tumor escape within immunotherapy and the immunodominant role of nonviral tumor antigens targeted by T cells. To avoid this, a strategy may be to administer peripheral blood T cells that are genetically engineered ex vivo, to target an HPV oncoprotein (E5, E6) with a T cell receptor [52].

ICIs have also demonstrated their potential efficacy in cervical cancer, reporting an ORR in recurrent/advanced disease ranging from 12.2 to 26% [53]. Whereas increased expression of PD-L1 on cervical tumors is associated with a better response to ICIs [54], the response rate in patients with PD-L1-negative is relatively poor [55]. Therefore, anti-PD-1 therapy alone may not be an effective treatment in metastatic cervical cancer patients who had low expression of PD-L1. As is known, the effects of antiPD-1 therapy are mediated by TILs in the tumor microenvironment, and therefore a combination of anti-PD-1 therapy and TILs may provide major antitumor effects. An interesting study reported the results of 80 PD-L1-negative cervical cancer patients treated with a TIL and anti-PD-1 combination therapy; the objective response was observed in 20 patients (25%), and the authors speculated that TILs penetrated the tumor microenvironment and secreted factors, which induced the expression of PD-L1 in the tumor cells [55].

Table 2 summarizes the selected ongoing clinical trials of ACT in virus-associated cancers.

### 2.4. T Cells Targeting Tumor Neoantigens

Unselected TILs also contain non-specific cells. Therefore, not all TIL products have potent antitumor effect, and patients with low levels of tumor antigens could have low levels of tumor-specific TILs. The identification and expansion of T cells with a specificity towards tumor antigens is an encouraging strategy for enhancing the chances of success of ACT. One of the most promising ways is to identify and expand the T cells with TCRs which are specific against tumor antigens. The genetic sequencing approach has permitted the recognition of neoantigens generated by gene rearrangements or mutations [56]. These neoantigens are expressed by all cells within a tumor, as well as across different tumor etiologies, and the identification of TCRs targeting these neoantigens may enable them to be utilized as specific reagents for TCR therapy [24]. Furthermore, neoantigens derived from somatic tumor mutations are likely to stimulate endogenous antitumor immunity, playing a critical role in maintaining durable responses after ACT [57]. Interesting results concerning the in vivo efficacy of T cells targeting tumor neoantigens have been provided in the setting of solid tumors. In particular, metastatic colon cancer patients, treated with TILs containing approximately 75% of CD8+ cells able to recognize a neoantigen derived from a hotspot mutation (G12D) in the KRAS gene, reported a regression of all metastatic lung lesions for almost 9 months [24]. TCRs against KRAS^G12D^ mutations restricted to HLA are now being tested in phase I–II of a clinical trial (NCT03745326).

Neoantigens have emerged as a major determinant of long-lasting tumor regressions in different immunotherapy trials; however, durable clinical responses are reported in only a minority of patients. One of the aspects that may negatively affect the treatment efficacy is the poor and heterogeneous expression of the antigens in the tumor load. Some strategies have been utilized to overcome these limitations: neoantigens should be identified in the patient’s tumor by tumor RNA sequencing. Secondly, bioinformatical approaches should be used to recognize the presence of the antigens in all cancer cells, and finally, targeting different neoantigens could minimize the tumor heterogeneity and also the chances of tumor resistance, due to the antigen loss or loss of specific MHC molecules [58].

## 3. Tumor Mutational Burden

Mutated genes can produce mutated proteins that can be identified by the immune system and presented on the cell surface by MHC molecules [58,59,60]. The tumor mutational burden (TMB) differs among distinct cancer types, and tumors with high TMB have high neoantigen expression, and therefore are predicted to be more immunogenic [61]. Previous experience has demonstrated that patients having a high TMB were more likely to respond to anti-PD1 therapy, in different solid tumor types [62]. Furthermore, tumors with deficiencies in the mismatch repair (MMR) pathway, have high levels of genetic mutations, and therefore high expression of neoantigens [63]. Clinical studies have reported that intestinal cancer patients with MMR deficiency are more likely to respond to ICIs as compared to patients with MMR-proficient tumors [64,65].

Tumors with high TMB and a consequently high expression of neoantigen have high levels of TIL clones among cancer cells, ensuring a sufficient number of tumor-reactive TILs in the final product. It is noteworthy that melanoma is well known for having a high TMB, and it has provided interesting results in TIL ACT.

Previous experiences have reported that TMB could be a biomarker of the response to immunotherapy in patients with different tumor etiologies treated with various types of immunotherapy [54]. Patients with a high TMB had a significantly higher response rate and longer progression-free survival than those with a lower TMB [54].

The association of improved anti-PD-1 and anti-PD-L1 clinical responses with high TMB strongly suggests also that neoantigens are an important target for antitumor immunity by PD-1 inhibition. Moreover, in the peripheral blood of advanced lung cancer patients who received anti-PD-1 therapy, there were detected neoantigen-specific CD8+ T cells, confirming the strong correlation between TMB and neoantigens [38].

## 4. Adaptive Cell Therapy with CAR-T: Progress and Pitfalls

Whilst TCR therapy requires HLA-restricted specificity and clinical efficacy depends on TCR affinity and the expression of MHC-antigen complexes on the tumor cells, CAR recognition of tumor target cells is HLA-unrestricted. CAR-T cells consist of an antigen-binding domain, derived from a single-chain variable fragment of a monoclonal antibody directed against a tumor-specific antigen, a hinge region, a trans-membrane domain, and the TCR intracellular domain, able to activate T cells [4]. The encouraging results obtained in hematological malignancies [66,67,68] have paved the way for the application of this approach in solid tumors, unfortunately with unsatisfactory results to date.

Despite ameliorating CAR construction, poor in vivo expansion and duration were reported in solid tumors [69,70,71,72]. Second- and third-generation CARs have included one and two costimulatory domains, respectively, aiming to increase cytotoxicity and cytokine production, and improve the proliferation and persistence of CAR-T cells [73,74]. More recently, fourth-generation CAR-T, which incorporate additional stimulatory domains, have been reported [75]. Fourth-generation CAR-T cells share the same construct as second-generation CARs, but include a protein such as interleukin-12 (IL-12) that is constitutively or inducibly expressed upon CAR activation, whilst fifth-generation CAR-T cells are based on the second generation of CARs, with the addition of a truncated cytoplasmic IL-2 receptor β-chain domain with a binding site for the transcription factor STAT3 [76].

Notwithstanding the poor evidence of the efficacy of CAR-T cells in solid tumors, interesting results have been published recently in the treatment of diffuse midline glioma with high expression of GD2 [77]. Along the same lines, a few weeks ago, the authors of [78] described noteworthy results in the treatment of high-risk advanced neuroblastoma with third-generation GD2-CAR-T cells expressing the inducible caspase 9 suicide gene (GD2-CART01). Among 27 heavily pretreated children, 9 patients reported a complete response and 8 patients had a partial response, with an overall response rate of 63%, and a 3-year overall survival and event-free survival of 60% and 36%, respectively [78]. This longer time for disease progression may be a consequence of a persistence of CAR-T cells in vivo, due to the incorporation of two costimulatory domains in the construct of GD2-CART01. The interesting findings reported in this study represent a challenge for immunotherapy in the treatment of solid tumors.

With the aim of increasing the efficacy of CAR-T, some authors have documented the fact that the spatial distance between CAR-T and the target antigens may be equally important for T cell signaling: CAR-T with a shortened extracellular spacer conferred superior recognition of tumor cells compared to the same scFv with a longer spacer [79]. Thus, the design of CAR-T for novel targets should factor in the location of the target epitope and customize the spacer length to optimize CAR-T cell signaling [79].

### 4.1. CAR-T Interactions with Tumor Microenvironment (TME)

The tumor microenvironment (TME) is a complex network that comprehends the extracellular matrix and several nonmalignant cells. Mechanisms involved in the immunosuppressive nature of TME include (i) physical barriers to tumor invasion by immune cells, (ii) upregulated checkpoint ligands, (iii) a pro-tumor stromal niche, (iv) immunosuppressive soluble factors, and (v) the selected expression of chemokines to recruit immunosuppressive cells such as regulatory T lymphocytes, M2 macrophages, myeloid-derived suppressor cells (MDSCs) and cancer-associated fibroblast (CAFs) (Figure 1) [75].

The efficacy of CAR-T cell therapy in solid cancers is drastically hampered by poor immune cell infiltration [80]. Tumor cells are able to regulate chemokines, contributing to the poor recruitment of CAR-T cells. Engineering CAR-T cells with receptors for chemokines that are overexpressed in the TME has been a challenge. Previous studies reported CAR-T cells engineered to co-express chemokine receptors (such as CCL2 or CCR4) able to enhance T cell homing and improve migration toward tumor cells [81,82]. Once CAR-T cells reach the tumor site, the high density of the extracellular matrix (ECM) associated with solid tumors is a hurdle to overcome. Because of this, some authors have engineered CAR-T cells to express heparanase, which is able to degrade ECM [83].

MDSCs contribute to creating an immunosuppressive environment enhancing the expansion and the activation of T-regs, CD4+ T lymphocytes that can inhibit the function of tumor-specific T lymphocytes producing immunosuppressive cytokines (IL-10 and TGF-β), expressing negative costimulatory molecules (CTLA-4, PD-1 and PD-L1) and consuming IL-2, a cytokine essential for the proliferation of cytotoxic T lymphocytes [84,85].

Lymphodepleting therapy, preceding CAR-T cell infusion, aims to eliminate the recipient’s lymphocyte pool, including T-regs, resulting in increased proliferation and expansion of reinfused T-lymphocytes and inhibition of the tumor immunotolerance mechanism [86].

CAFs are a great part of the tumor stroma, and CXCL12 secreted by CAFs plays an important role in ‘protecting’ tumor cells from immune attack and keeping T cells out of the tumor microenvironment. The fibroblast activation protein (FAP) is highly expressed in CAFs in over 90% of solid tumors, making this protein a potential therapeutic target. Encouraging results have emerged from several studies conducted in vitro and in vivo on CAR-T cells, directed against FAP [87]. Moreover, the aberrant tumor vasculature causes interstitial hypertension that prevents extravasation and a hypoxic microenvironment, especially in the central part of the tumor. Thus, normalizing the tumor vasculature may be beneficial, and anti-vascular endothelial growth factor receptor (VEGFR) CAR-T cells may play a role in the inhibition of the tumor growth [88]. Considering the metabolic barriers, it is worth noticing that the particular anatomical structure of solid tumors generates hostile hypoxia, able to hamper the expansion of CAR-T cells. Strategies to favor CAR-T response in the hypoxic TME are under investigation [89].

Other strategies are studied to improve the efficacy of CAR-T cells. Some cytokines play a role in enhancing CAR-T cell proliferation, survival, and effector function in the immunosuppressive TME. Most cytokines studied in this setting are IL2, IL7, IL15 or IL21, and their association with CAR-T cells has been described in vivo [90].

### 4.2. CAR-T: How to Increase Endogenous Immune Response

It is well known that tumor cells are able to escape subsets of leukocytes, including CAR-T cells, and to induce immunosuppressive phenotypes [91]. Recently, T cells engineered to secrete Fms-like tyrosine kinase 3 ligand (Flt3L), a hematopoietic cell growth factor, have been able to promote intratumoral proliferation of dendritic cells (DCs), and particularly to induce immunity against tumor cells [92].

Furthermore, CAR-T cells have also been engineered to express different immune-modulatory proteins. For example, CD40L is expressed on T cells, and its interaction with its receptor can lead to the activation of APCs and the apoptosis of CD40+ tumor cells [75]. CD40L expression on CAR-T cells resulted in elevated surface expression of costimulatory molecules, adhesion molecules, HLA molecules and Fas death receptor on CD40+ tumor cells, thus increasing their immunogenicity [92]. These T cells also induced the secretion of proinflammatory IL-12 and showed enhanced antitumor activity in vitro and in vivo [93]. Finally, CAR-T cells can be engineered in molecules able to engage tumor cells by endogenous, non-engineered T cells [94].

“Exhaustion” is a state of functional hyporesponsiveness of T cells during chronic infections, which arises when pathogens are not quickly eliminated, but rather persist. It has been suggested that T cells, and also CAR- and TCR-T cells, exhibit an “exhausted” state in the contest of solid tumors, similar to chronic infections, due to the high antigen load and the immunosuppressive factors in the TME. Recent experiences have documented an enrichment of exhaustion genes and TCR pathways, suggesting that the T cell unresponsiveness is also driven by antigen chronicity and duration of TCR stimulation [95]. Thus, tumor-induced T cell dysfunction is driven by the presence and persistence of tumor antigens, and although microenvironmental factors may contribute, they are not sufficient to induce the dysfunctional phenotype [95,96]. Potential transcription factors able to control the dysregulated gene expression signature are also involved [96]. Therefore, strategies to overcome this antigen-driven dysfunction may be required to improve cancer immunotherapy.

Table 3 summarizes selected ongoing clinical trials of CAR-T therapy.

## 5. Conclusions

Recent cell engineering strategies have made great strides in ameliorating the antitumor function of T cells, increasing tumor-targeting specificity, preventing tumor escape, and modifying the TME. Furthermore, the progress in next-generation sequencing and other analyses could enhance our ability to understand these complex interactions and to create the next generation of T cell therapy.

## Figures and Tables

**Figure 1 cancers-15-03667-f001:**
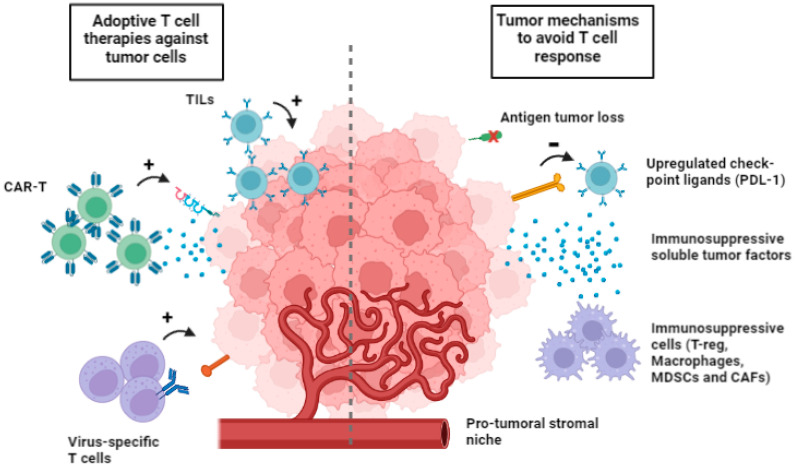
Multiple cell types available for adoptive cell therapies (on the **left**) and tumor mechanism to avoid T cell response (on the **right**). Tumor evasion includes loss of tumor neoantigens, upregulation of checkpoint ligands, immunosuppressive soluble tumor factors, immunosuppressive cells, such as T-reg, macrophages, myeloid-derived suppressor cells (MDSCs) and cancer-associated fibroblast (CAFs).

**Table 1 cancers-15-03667-t001:** Selected ongoing clinical trials of ATC therapy with T cells, recruiting advanced cancer patients other than melanoma (clinicaltrials.gov).

Interventions	Disease	Phase Clinical Trial	Treatments	Endpoints	NCT
Autologous TILs (LN-145)	Metastatic NSCLC	Phase II	Lymphodepleting chemotherapy with Flu and Cy, prior to TIL infusion	ORR	NCT04614103
TILs in combination with immunotherapy	Metastatic OC	Phase I-II	Lymphodepleting chemotherapy with Flu and Cy, prior to TIL infusion; subsequent therapy with Nivolumab and Relatlimab; a separate group of patients will also receive Ipilimumab before lymphodepleting chemotherapy	Safety ORR	NCT04611126
Adoptive T cell therapy	Metastatic urothelial carcinoma	Phase II	TILs after lymphodepleting chemotherapy; subsequently high-dose bolus IL2	ORR	NCT04383067
Autologous TILs (LN-145)	Metastatic TNBC	Phase II	TILs (LN-145) after lymphodepleting chemotherapy; subsequently IL2	ORR	NCT04111510
Adoptive T cell therapy	Recurrent OC	Phase I-II	TILs after 6 cycles of carboplatin/taxol; a separate group will also receive Interferon Alfa2A	Safety ORR	NCT04072263
Adoptive T cell therapy	Solid tumors	Phase II	TILs after lymphodepleting chemotherapy with Flu and Cy	ORR	NCT03935893
Adoptive T cell therapy	Biliary carcinoma; Cholangiocarcinoma	Phase II	TILs and Adesleukin	ORR	NCT03801083
TILs alone or in combination with immunotherapy	Metastatic Melanoma or NSCLC or SCC Head and Neck	Phase II	LN-144 or LN-145 or LN-145-S1 alone or in combination with Ipilimumab or with Nivolumab or with Pembrolizumab	ORR	NCT03645928
Autologous TILs (LN-145 or LN-145-S1)	Solid Tumors	Phase II	TILs after lymphodepleting chemotherapy (Flu and Cy), plus Adesleukin, alone or in combination with Ipilimumab/Nivolumab (according to histology)	ORR	NCT03449108
Adoptive T cell therapy	Cervical cancer	Phase II	TILs (LN-145) after lymphodepleting chemotherapy followed by IL2, alone or in combination with Pembrolizumab	ORR	NCT03108495
Adoptive T cell therapy	NSCLC	Phase II	TILs (Young TIL) after lymphodepleting chemotherapy followed by high-dose Adesleukin or low-dose Adesleukin	ORR	NCT02133196
Adoptive T cell therapy in combination with Pembrolizumab	Solid Tumors	Phase II	TILs (Young TIL) after lymphodepleting chemotherapy followed by high-dose Adesleukin plus Pembrolizumab	ORR	NCT01174121

Abbreviation: NSCLC = non-small-cell lung cancer; Flu = Fludarabine; Cy = Cyclophosphamide; ORR = objective response rate; OC = ovarian cancer; TNBC = triple-negative breast cancer; SCC = squamous-cell carcinoma.

**Table 2 cancers-15-03667-t002:** Selected ongoing clinical trials of ACT therapy in virus-associated cancers (clinicaltrials.gov).

Interventions	Disease	Phase Clinical Trial	Treatments	Endpoints	NCT
TCR-engineered T cells targeting HPV-16 E7	HPV+ refractory carcinoma	Phase I-II	Cy and Flu preparative regimen followed by TCR(E7) T cells; high-dose aldesleukin	Safety and ORR	NCT02858310
E7 TCR-T cell induction therapy for advanced HPV-related tumors	HPV-related tumors	Phase I-II	E7 TCR-T cells; aldesleukin	Safety and ORR	NCT05639972
TG4001 and Avelumab in HPV16positive R/M cancers	HPV-related tumors	Phase Ib/II	TG4001; Avelumab	Safety and ORR	NCT03260023
A vaccine (PDS0101) alone or in combination with Pembrolizumab for locally advanced HPV-related oropharynx cancer	Oropharyngeal carcinoma (Stage III or IV); HPV-related disease	Phase I/II	Liposomal HPV-16 E6/E7 multipeptide vaccine PDS0101; Pembrolizumab	Pathologic and ctHPVDNA response; ORR	NCT05232851
HPV-16/18 E6/E7 specific T lymphocytes relapsed HPV-associated cancers	HPV-related cancers	Phase I	Cy and Flu lymphodepleting regimen followed by HPV-specific T cells (Group A); if safe, additional group of patients (Group B) also receive Nivolumab	Safety; ORR	NCT02379520
R/M HPV-16 related SCC	HPV-related SCC	Phase I/II	TheraT^®^ Vectors expressing HPV 16+ specific antigens (drugs: HB-201 and HB-202)	Safety; ORR	NCT04180215
HPV-16 vaccination and Pembrolizumab plus CTRT for HPV-16-related head and neck SCC	Locally advanced, intermediate risk, HPV-associated head and neck SCC	Phase II	Combination of Pembrolizumab, HPV-16 E6/E7 specific vaccination (ISA101b) and cisplatin-based chemoradiotherapy	PFS at 2 years	NCT04369937

Abbreviation: TCR = T cell receptor; Cy = cyclophosphamide; Flu = fludarabine; HPV= human papilloma virus; ORR = overall response rate; R/M = refractory/metastatic; ctHPVDNA = human papillomavirus cell-free tumor deoxyribonucleic acid; SCC = squamous cell carcinoma; CTRT = chemoradiotherapy; PFS = progression-free survival.

**Table 3 cancers-15-03667-t003:** Selected ongoing clinical trials of CAR-T therapy in solid tumors (clinicaltrials.gov).

Interventions	Disease	Phase Clinical Trial	Treatments	End Points	NCT
HLA-G- targeted CAR-T cells IVS-3001	Advanced HLA-G+ solid tumors	Phase I/IIa	Cy and Flu preparative regimen followed by IVS-3001 Anti-HLA-G CAR-T	Safety; clinical activity	NCT05672459
HER2 targeted HypoSti.CAR-T cells	HER2+ advanced solid tumors	Phase I/II	Preconditioning regimen of albumin-bound paclitaxel and Cy, before HypoSti.CAR-HER2 T cells	Safety; ORR	NCT05681650
NKG2D-based CAR-T cells in patients with NKG2DL+ solid tumors	HCC; GBL; medulloblastoma; colon cancer	Phase I	NKG2D-based CAR-T cells	Safety	NCT05131763
B7H3 CAR-T cells in advanced solid tumors in children and AYAs	STS; EWS; GCC; OS; pediatric solid tumors	Phase I	B7H3-EGFRt- DHFR	Safety; tolerability	NCT04483778
EGFR806 CAR-T cells in advanced solid tumors in children and AYAs	STS; EWS; OS; GCC; pediatric solid tumors	Phase I	EGFR806	Maximum tolerated dose; DLT	NCT03618381
GD2/PSMA Bi-specific CAR-T cells in advanced solid tumors	GD2 and PSMA positive solid tumors	Phase I/II	bi-4SCAR GD2/PSMA T cells	Safety; ORR	NCT05437315
CLDN 18.2 targeting CAR-T cells in advanced solid tumors	Gastric cancer, pancreatic cancer, and ovarian cancer with CLDN 18.2 expression	Phase I	Claudin 18.2 CAR-T	Safety; ORR	NCT05472857
LYL797, ROR1-targeting CAR-T cells in adult refractory solid tumors	TNBC; NSCLC	Phase I	LYL797	Safety; ORR	NCT05274451
HER2- targeted dual switch CAR-T cells in advanced HER2-positive cancer	HER2+ gene amplification	Phase I/II	HER2-targeted dual-switch CAR-T cells	Safety; ORR	NCT04650451
Binary Oncolytic Adenovirus in combination with HER2-specific CAR-T cells in patients with advanced HER2+ solid tumors	Bladder cancer; Breast cancer; Lung cancer; HNSCC; Gastric cancer; Colorectal cancer; Pancreatic cancer.	Phase I	CAdVEC	DLT; ORR	NCT03740256
CAR-T-targeting CEA in the treatment of CEA positive advanced solid tumors	Gastric cancer; Colon cancer; Rectal cancer; Esophageal cancer; Pancreatic cancer.	Phase I/II	Cy and Flu preparative regimen followed by CEA-targeting CAR-T cells	Safety; ORR	NCT05538195
IL-15 armored Glypican-3-specific CAR-T cells in pediatric solid tumors	Liver cancer; Liposarcoma; Wilms tumor; Yolk sac tumor; Rhabdomyosarcoma.	Phase I	Cy and Flu preparative regimen followed by AGAR T cells (GPC3-CAR and the IL15)	DLT; ORR	NCT04377932

Abbreviation: Cy = cyclophosphamide; Flu = fludarabine; ORR = overall response rate; HCC = hepatocellular carcinoma; GBL = glioblastoma; AYAs = adolescent and young adults; STS = soft tissue sarcoma; EWS = Ewing sarcoma; GCC = germ cell cancer; OS = osteosarcoma; DLT = dose-limiting toxicities; BOR = best overall response; TNBC = triple-negative breast cancer; NSCLC = non-small-cell lung cancer; HNSCC = head and neck squamous cell carcinoma.

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
