# Peer review of "Clinical Trials of Cellular Therapies in Solid Tumors"

_cancers, 2023, doi:10.3390/cancers15143667_

Round 1
Reviewer 1 Report
With the current manuscript, Secondino and colleagues offer an overview of the current state of clinical trials using cellular therapies in solid tumors. The review is well-written and informative, and most references are up to date. After a brief introduction, the authors offer an outline of adaptive immunotherapy by expansion of T cells in vitro, then they add an interesting section on the effects of tumor mutational burden on immunotherapy and they close with a brief section on CAR-T. Although the paper is well annotated, there are still some missing references and points that I would suggest to address
- In the section of CAR T therapy in solid tumors, I would slightly expand the design concepts [see Srivastava and Riddell - Engineering CAR-T cells: Design concepts – Trends Immunol 2015]. Also, while mentioning the progress and pitfalls of CAR T therapy in solid tumors, I would briefly mention biochemical, physical and metabolic barriers that hamper CAR T efficacy (see Boccalatte et al –Advances and hurdles of CAR T immune therapy in solid tumors – Cancers 2022)
- The GD2 CAR T trial cannot omit mentioning [Majzner et al – Nature 2022]
- To better describe the states of T cell exhaustion, I would cite [Schietinger et al – Tumor-Specific T Cell Dysfunction Is a Dynamic Antigen-Driven Differentiation Program Initiated Early during Tumorigenesis - Immunity 2016]. It would also be interesting to show some mechanistic characterizations of exhaustion [Good, Aznar et al An NK-like CAR T cell transition in CAR T cell dysfunction – Cell 2021]
Author Response
I'd like to thank the Reviewer for his comment.
I have modified the text according to the Reviewer's suggestions: I added the References (Srivastava and Riddell, 2015; Boccalatte 2022; Majzne, 2022; Schietinger, 2016 and Good, 2021) trying to summarize them in the text.
Reviewer 2 Report
This manuscript by Secondino et al aims at providing an overview on current adoptive T-cell therapy approaches in solid tumors, research performed to increase their efficacy and safety, and summarizing results from ongoing clinical trials. In general, the authors achieve these goals, although not much is mentioned about safety.
The authors have extensive experience in the field and provide an important insight on the historical evolution of cellular immunotherapies, and on the current challenges in the treatment of solid tumors. The manuscript is well redacted, however there are some grammatical issues that should be improved. Moreover, the authors refer to adoptive cellular therapies here and there as to “adaptive”. This should be corrected.
The selection of the studies mentioned is forcedly limited, but e.g.
L234 NeoTIL approach could be mentioned. E.g. NCT04643574 NeoTIL in Advanced Solid Tumors
and
L363 Table 3: study GD2-CART01 NCT03373097 has published results, it should be mentioned. 10.1056/NEJMoa2210859
Minor comments
L15 Adoptive cell therapies (not adaptive)
L21 will likely play
L22 cancer treatments have
...check grammar...
L314 Figure 1: text red underline – remove, Abbreviation: explain in legend (e.g. MDSC...)
L291 to mention only scFv as ABD is a limitation. Other ABD are being tested.
L321 “Tumor cells are able to regulate chemokines” not clear what this means
L328 Heparanase
The manuscript is well redacted, however there are some grammatical issues that should be improved. Moreover, the authors refer to adoptive cellular therapies here and there as to “adaptive”. This should be corrected.
Author Response
First of all, thanks to the Reviewer for his comments.
I have corrected the minor comments, as suggested.
Among the studies mentioned in table 1, we have decided to mention only phase II or phase I-II trials (NCT04643574 NeoTIL in Advanced Solid Tumors is only a phase I study).
We have modified the table 3, and the study GD2-CART01 NCT03373097 has been mentioned in the text.